# Risk of negative birth experience in trial of labor after cesarean delivery: A population-based cohort study

**Charlotte Lindblad Wollmann**[1,2]*, **Can Liu**[1], **Sissel Saltvedt**[2], **Charlotte Elvander**[1], **Mia Ahlberg**[1,2], **Olof Stephansson**[1,2]

**1** Clinical Epidemiology Division, Department of Medicine, Karolinska University Hospital and Institutet, Stockholm, Sweden, **2** Division of Obstetrics and Gynecology, Department of Women´s and Children´s Health, Karolinska University Hospital and Institutet, Stockholm, Sweden

* charlotte.lindblad-wollmann@sll.se

**Data Availability Statement:** The data belong to "Graviditetsregistret" in Sweden, which does not permit data-sharing. Interested researchers who meet the criteria for access to confidential

## Abstract

### Background

To improve care for women going through trial of labor after cesarean (TOLAC), we need to understand their birth experience better. We investigated the association between mode of delivery on birth experience in second birth among women with a first cesarean.

### Methods

A population-based cohort study based on the Swedish Pregnancy Register with 808 women with a first cesarean and eligible for TOLAC in 2014–2017. Outcomes were mean birth experience measured by visual analogue scale (VAS) score from 1–10 and having a negative birth experience defined as VAS score ≤5. Linear and logistic regression analyses were performed with β-estimates and odds ratios (OR) with 95% confidence intervals (CI).

### Results

Mean VAS score among women with an elective repeat cesarean (n = 251 (31%)), vaginal birth (n = 388 (48%)) or unplanned repeat cesarean (n = 169 (21%)) in second birth were 8.8 (standard deviation SD 1.4), 8.0 (SD 2.0) and 7.6 (SD 2.1), respectively. Compared to women having an elective repeat cesarean, women having an unplanned repeat cesarean delivery had five-fold higher odds of negative birth experience (adjusted OR 5.0, 95% CI 1.5–16.5). Women having a first elective cesarean and a subsequent unplanned repeat cesarean delivery had the highest odds of negative birth experience (crude OR 7.3, 95% CI 1.5–35.5).

### Conclusions

Most women with a first cesarean scored their second birth experience as positive irrespective of mode of delivery. However, the odds of a negative birth experience increased among women having an unplanned repeat cesarean delivery, especially when the first cesarean delivery was elective.

information may request data from info@graviditetsregistret.se. The researchers behind this study has requested data from Graviditetsregistret without any privileges. Request for data on an aggregated level may be sent to Olof Stephansson at the Clinical Epidemiology Division, Department of Medicine, Solna, Karolinska University Hospital and Institutet, Stockholm, Sweden, olof.stephansson@ki.se.

**Funding:** This study was supported by grants from the Swedish Research Council (2013-2429, OS) (https://www.vr.se/english.html) and by grants provided by the Stockholm County Council (ALF project 2017-0189, OS)(https://forskningsstod. vmi.se/Ansokan/start.asp). The founding sources had no role in study design, collection of data, analysis or interpretation of data, nor in decision to submit the article for publication.

**Competing interests:** The authors have declared that no competing interests exist.

**Abbreviations:** CD, Cesarean Delivery; CI, Confidence Interval; ERCD, Elective Repeat Cesarean Delivery; OR, Odds Ratio; TOLAC, Trial Of Labor After Cesarean; URCD, Unplanned Repeat Cesarean Delivery; VAS, Visual Analogue Scale; VBAC, Vaginal Birth After Cesarean.

## Introduction

Childbirth is an important life event in women´s life. Anticipating triumph and delight, most women accept the possible difficulties of labor as part of a process to achieve a positive outcome for them and their child [1]. One in 10 have a negative birth experience which can affect the everyday life of the woman and her family, impair bonding with the newborn, even prolong birth intervals and impair future fertility [2–5]. A positive childbirth experience is an important aspect of intrapartum care and is highlighted in the new World Health Organization (WHO) guideline [6].

Birth experience is multidimensional, affected by maternal age, fear of childbirth, support from the midwife and partner during labor, induction of labor, labor duration, pain, expectations of giving birth, involvement and participation during labor, and surgical procedures [5, 7–13]. Mode of delivery, often affected by the factors listed above, is naturally an important indicator for the birth experience. Among first time mothers, having a vaginal non-instrumental delivery is associated with the highest rated birth experience, whereas unplanned cesarean delivery (CD) is associated with worse birth experience [5, 7, 14, 15].

The rate of CD is rising globally. Having a previous CD, as a reason for another CD, further increases CD rates [16, 17]. Therefore, a trial of labor after one cesarean (TOLAC) is promoted in many countries to lower CD rates and associated maternal morbidities [18–20]. Nonetheless, TOLAC may bare the risk of unplanned repeat CD with its associated risk of adverse medical outcomes [20, 21].

The knowledge about birth experience in subsequent delivery after a previous CD is sparse [22–25]. Previous studies were furthermore hampered by confounding, had small or unrepresentative sampling, and thereby had limited validity and generalizability [22–25]. To fill the knowledge gap of birth experience among women with a first CD and to improve the quality of care, we aim to study the impact of the second mode of delivery on birth experience in women with a first CD.

## Methods

### Study population

The Swedish Pregnancy Register, founded in 2013, included in 2017 approximately 93% of all births in Sweden [26]. The register contains detailed data on the pregnant women prospectively entered into the electronic medical records by midwives and physicians in a standardized way at first antenatal visit and at every subsequent visit, ultrasound examinations, delivery and postnatal care [26]. After delivery, before discharge from hospital, women giving birth in Sweden are in many hospitals asked about their birth experience by using a visual analogue scale (VAS) scoring from 1 to 10, where 10 is a very positive and 1 is a very negative birth experience. The midwife responsible of the woman´s postnatal care mainly asks the question about birth experience. It is asked either as an oral question or through a questionnaire, varying at different maternity units. The individual VAS scores are entered in the electronic medical records by the responsible midwife and forwarded into the Swedish Pregnancy Register.

All Swedish citizens and immigrants with one-year or longer residence receive their unique personal identification number at birth or immigration. This together with our nationwide register enables a unique possibility for longitudinal research [27]. All pregnant women are offered free maternity care in Sweden and the insurance system does not influence the availability of this care. More than 98% of pregnant women participate in the antenatal care system and more than 99% of all births take place in hospitals. The hospitals range between 1, 2 and 3 level hospitals and are mainly public. [28] There are no units lead by only midwifes.

In this study, we included all women registered in the Swedish Pregnancy Register with a first cesarean delivery and a subsequent birth to a singleton, live-born infant in a cephalic presentation at or above 37 gestational weeks during 2014–2017. Women were classified as eligible for TOLAC when the presentation was cephalic and there was no placenta previa or other medical contraindication for trial of labor. The preliminary analysis showed that some regions had a lower rate of reporting birth experience, possibly due to differences in the evaluation of the VAS scale as an instrument to measure birth experience, however response rates did increase over time. To diminish confounding by organizational factors, we chose arbitrarily to exclude women giving birth in hospitals with a birth experience response rate of less than 80% in 2017 (excluded hospitals n = 23). A few hospitals (n = 7) in the southeast region of Sweden had until 2017 the opposite interpretation of the VAS score, and these hospitals were excluded since it was unclear the exact time point for reversing the VAS scale. We also excluded births with missing birth experience data, and a sensitivity analysis was performed comparing excluded and included women. In the end, the national sample gave us a population of 808 women with first and second birth in either of the remaining 12 hospitals (Fig 1). These remaining hospitals are widespread over Sweden, including a range of university clinics to smaller country-based clinics.

## Exposure

The main exposure of interest was mode of delivery in 2nd birth categorized into: 1) elective repeat cesarean delivery (ERCD), 2) vaginal birth after cesarean (VBAC) or 3) unplanned repeat cesarean delivery. Women with an ERCD were used as the reference. In a supplementary analysis, we also studied intended mode of delivery (TOLAC or ERCD) as the exposure.

When studying 1st and 2nd mode of delivery and birth experience, we further categorized the exposure of interest by the 1st mode of delivery, categorized into A) elective CD or B) unplanned CD. Women, where both 1st and 2nd birth was elective CD, were used as the reference.

## Outcome

Mean birth experience VAS score in the 2nd delivery was the main outcome. Previous studies have shown that about 10% of women assessed their birth as negative when using a scoring tool similar to a VAS [5]. The tenth percentile of the distribution of birth experience was 5 in our study. Thereby, to find women scoring their birth experience as negative, we dichotomized the VAS score and defined negative birth experience as VAS score $\leq 5$.

## Covariates

Based on previous studies and clinical experience, we considered the following covariates as confounders. The covariates were adjusted for in a step-wise regression analysis. In Model 1 adjustments were made for maternal characteristics including maternal age at 2nd birth, body mass index (BMI), height and cohabiting at first antenatal visit in the 2nd pregnancy, education ($\leq 9$ years of basic education, secondary school, university or college education) and self-assessed health at early 2nd pregnancy (categorized from very bad to very good). In Model 2 we adjusted for the same covariates as in Model 1 and the women´s previous childbirth experiences such as fear of childbirth in 2nd pregnancy (extra support from either midwife, obstetrician or psychologist), birth experience in 1st birth (measured through VAS score) and additionally, mode of delivery in 1st birth (elective or unplanned CD).

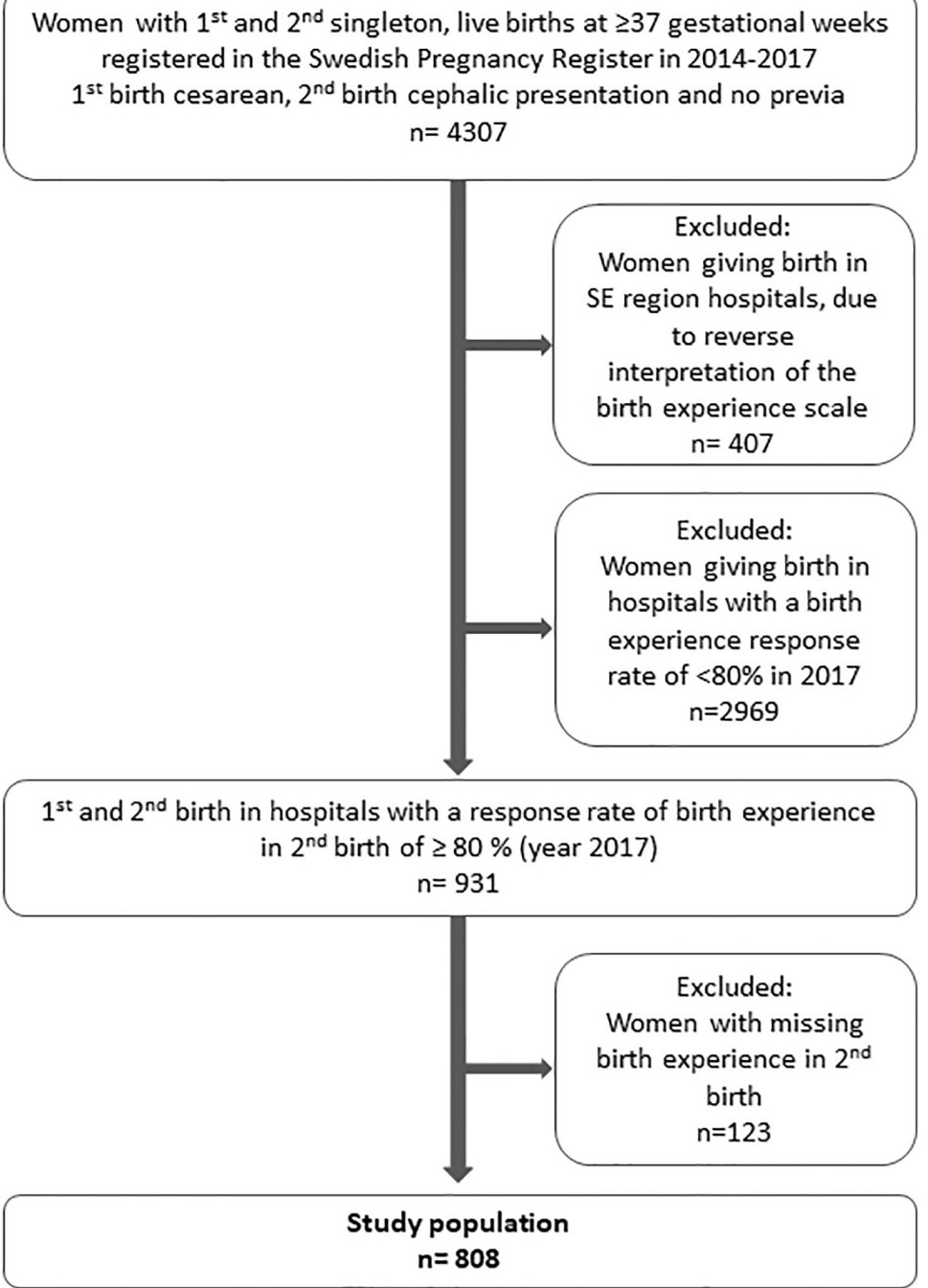

**Fig 1. Flowchart of the study population.**

## Statistical analyses

Maternal characteristics by mode of 2nd delivery were analyzed using the Chi-square test and Student´s t-test. Since the normality assumption of linear regression was not violated, we calculated the mean differences of birth experience (β-coefficient) through linear regression analysis.

With logistic regression models, we calculated odds ratios (OR) for negative birth experience. We also performed analysis for mean birth experience and ORs for negative birth experience by 1st and 2nd mode of delivery. We investigated possible effect modification between mode of delivery in 1st and in 2nd birth associated with the birth experience. We performed a sensitivity analysis for mean VAS scores including all hospitals, except the Southeast region. Finally, a sensitivity analysis was performed comparing women giving birth in the included hospitals with and without birth experience score. A two-sided p-value <0.05 was considered statistically significant. Statistical analyses were performed using the SAS software version 9.4.

### Ethical approval

In this study, there were no requirement of informed consent from the women studied. The results are presented on an aggregated level and all data was fully anonymized in the research database. The regional ethical committee at Karolinska Institutet, Stockholm, Sweden approved the study protocol (No 2017/2385-31/5 and No 2018/601-32).

### Results

Out of the 808 included women, 69% (n = 557) had a TOLAC and 31% (n = 251) delivered by ERCD. TOLAC rates varied between 60–91% in the included hospitals. Of the women undergoing TOLAC 70% (388) had a vaginal birth (VBAC) and in 30% (n = 169) the delivery ended with an unplanned repeat cesarean. Compared with the TOLAC group, women in the ERCD group were generally older, had a higher level of education, had lower gestational age, and were more likely to have fear of childbirth (S1 Table).

Women in the VBAC group were generally younger, had lower education and their prevalence of fear of childbirth was lower in comparison with women with an ERCD (Table 1).

The distribution of birth experience in 2nd birth by mode of delivery was skewed towards higher numbers, as shown in Fig 2. More than 60% of all women, independent of mode of delivery in 2nd birth, scored a birth experience of 8 or more (ERCD 84.5%, VBAC 73.2% and unplanned repeat CD 63.3%, respectively).

The mean birth experience in women with an ERCD was 8.8. After adjusting for confounders, women with VBAC and unplanned repeat CD had 0.5 (95% CI; -0.9 to 0.01) and 0.9 (95% CI; -1.4 to -0.3) lower mean difference (β) of birth experience in comparison with women with an ERCD (Table 2). A similar comparison between ERCD and intended mode of delivery (TOLAC) is presented in S2 Table.

Women giving birth by unplanned repeat CD had an increased odds of a negative birth experience (aOR 5.0, 95% CI; 1.5 to 16.5) in comparison with ERCD (Table 3). A similar comparison between ERCD and TOLAC is shown in S3 Table.

When further studying 1st mode of delivery, it seems that women with a first elective cesarean and a second unplanned repeat CD had the highest odds of negative birth experience (crude OR 7.3, 95% CI; 1.5–35.5) (Table 4). When investigating the possibility of effect modification by mode of delivery in 1st birth (elective vs unplanned CD), the overall interaction term was non-significant (p = 0.65).

When including all hospitals in the Swedish Pregnancy Register, except the Southeast region with reverse interpretation of the VAS-scale, our sensitivity analysis showed little discrepancy from the results shown in Table 2 (mean VAS in ERCD 8.8, SD 1.4; VBAC 7.9, SD 2.0; URCD 7.5, SD 2.3). Finally, in the hospitals included women with missing data for birth experience did not significantly differ in respect of maternal characteristics and mode of delivery (S4 Table).

**Table 1. Maternal characteristics by mode of delivery in 2nd birth.**

| Characteristics 2nd pregnancy | | ERCD[a] | | VBAC[b] | | [d]p-value | Unplanned repeat CD[c] | | [d]p-value |
|---|---|---|---|---|---|---|---|---|---|
| n = 808 | | | | | | | | | |
| (n, %) | | 251 | 31.1 | 388 | 48.0 | | 169 | 20.9 | |
| **Demographics** (Mean ±SD) | | | | | | | | | |
| Age | | 33.2 | 4.9 | 31.0 | 4.3 | <0.0001 | 32.2 | 4.4 | 0.03 |
| Height | | 165.6 | 6.8 | 166.1 | 6.2 | 0.43 | 163.6 | 6.6 | 0.003 |
| BMI | | 25.6 | 5.2 | 24.8 | 4.6 | 0.06 | 27.4 | 5.7 | 0.0009 |
| **Social** (n, %) | | | | | | | | | |
| Cohabiting | | 240 | 95.6 | 379 | 97.7 | 0.14 | 156 | 92.3 | 0.15 |
| Smoker in early pregnancy | | 5 | 2.0 | 11 | 2.8 | 0.65 | 8 | 4.7 | 0.02 |
| Alcohol Audit >6 | | 1 | 0.4 | 2 | 0.5 | 0.95 | 0 | 0 | 0.67 |
| **Health** (n, %) | | | | | | | | | |
| Received care for mental health issues | | 15 | 6.0 | 26 | 6.7 | 0.07 | 5 | 3.0 | 0.13 |
| Self-assessed health at early pregnancy | | | | | | 0.06 | | | 0.70 |
| *Very poor or poor* | | 3 | 1.2 | 11 | 2.8 | | 2 | 1.2 | |
| *Neither poor or good* | | 14 | 5.6 | 11 | 2.8 | | 11 | 6.5 | |
| *Good or very good* | | 201 | 80.1 | 330 | 85.1 | | 140 | 82.8 | |
| *Missing or don´t know* | | 33 | 13.2 | 36 | 9.3 | | 16 | 9.5 | |
| **Education** (n, %) | | | | | | 0.01 | | | 0.04 |
| *≤9 years basic education* | | 7 | 2.8 | 24 | 6.2 | | 14 | 8.3 | |
| *Secondary school education* | | 70 | 27.9 | 142 | 36.6 | | 55 | 32.5 | |
| *University and college education* | | 138 | 55.0 | 176 | 45.4 | | 81 | 47.9 | |
| *Missing or unknown* | | 36 | 14.3 | 46 | 11.9 | | 19 | 11.2 | |
| **Birth experience 1st birth** | | | | | | | | | |
| *(Mean ±SD)* | | 7.6 | 2.3 | 7.6 | 2.2 | 0.94 | 7.3 | 2.4 | 0.22 |
| *(Median and IQR)* | | 8.0 | 3.0 | 8.0 | 2.0 | | 8.0 | 3.0 | |
| *Missing* | | 78 | 31.1 | 151 | 38.9 | | 64 | 37.9 | |
| **Fear of childbirth in 2nd pregnancy** (n, %) | | 97 | 38.7 | 75 | 19.3 | <0.0001 | 60 | 35.5 | 0.18 |
| **Gestational age in weeks** (Mean ±SD) | | 38.6 | 0.7 | 39.7 | 1.2 | <0.0001 | 39.6 | 1.4 | <0.0001 |

[a]ERCD, elective repeat cesarean delivery;

[b]VBAC, vaginal birth after cesarean;

[c]CD, cesarean delivery;

[d]p-values calculated in comparison to ERCD

# Discussion

In this population-based cohort study, we found an association between mode of delivery and birth experience among women with a previous CD. Most women with a previous cesarean scored their birth experience as positive independent of second mode of delivery. Women having a vaginal second birth had on average a slightly lower mean VAS score. Women with an unplanned repeat CD had the lowest mean VAS score and a five-fold increased odds of negative birth experience in comparison with women with an ERCD. It seems that women who had an elective CD in first and unplanned CD in second delivery were at highest risk for having a negative birth experience.

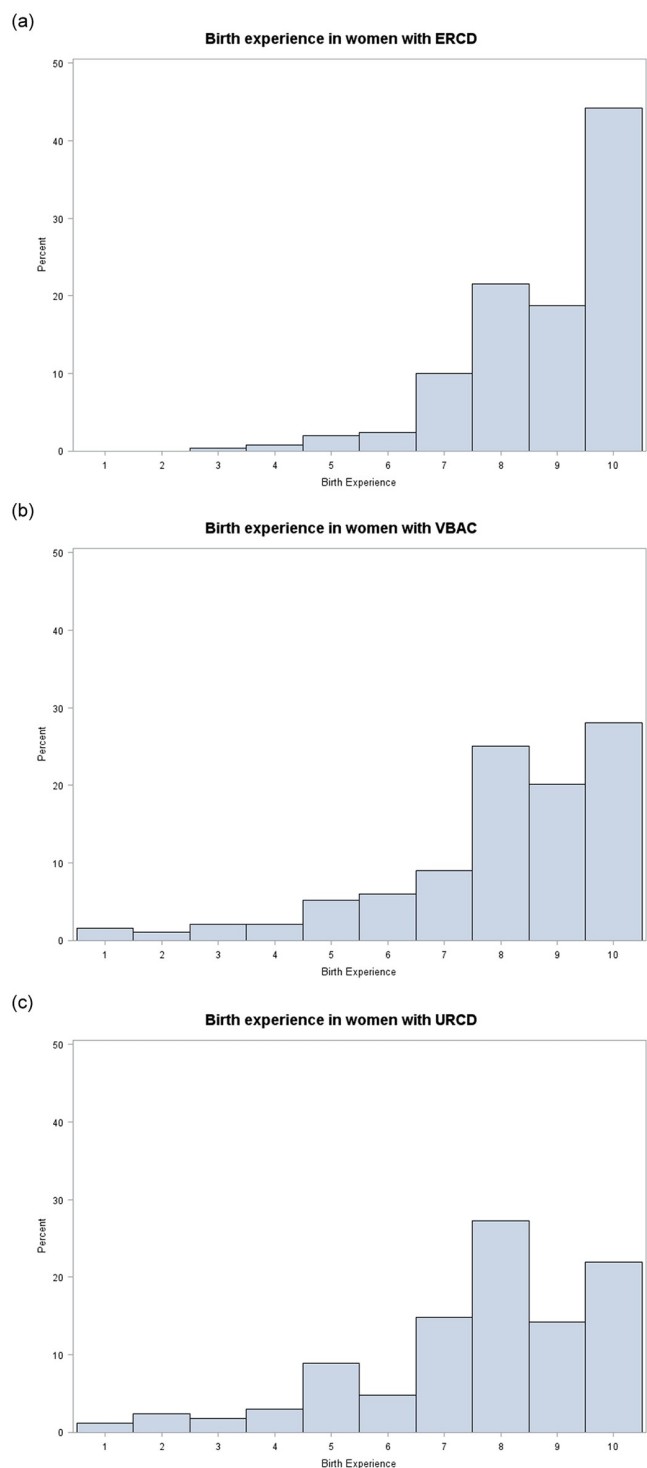

**Fig 2. Histogram of birth experience after 2<sup>nd</sup> birth by mode of delivery in 2<sup>nd</sup> birth; Elective repeat cesarean delivery (ERCD), vaginal birth after cesarean (VBAC) and unplanned repeat cesarean delivery (URCD).**

**Table 2. Mean difference of birth experience by mode of delivery in 2nd birth, linear regression.**

| Mode of delivery in 2nd birth[b] | Birth experience 2nd birth | | | | | | |
|---|---|---|---|---|---|---|---|
| | Crude | | | Model 1[a] | | Model 2[a] | |
| | Mean (SD) | β | 95% CI | β | 95% CI | β | 95% CI |
| ERCD | 8.8 (1.4) | Reference | | Reference | | Reference | |
| VBAC | 8.0 (2.0) | -0.8 | -1.1 to -0.5 | -0.8 | -1.1 to -0.4 | -0.5 | -0.9 to 0.01 |
| URCD | 7.6 (2.1) | -1.2 | -1.5 to -0.8 | -1.1 | -1.5 to -0.7 | -0.9 | -1.4 to -0.3 |

[a]Adjustment in

Model 1: maternal age, height, BMI, cohabiting, education, self-assessed health in 2nd pregnancy

Model 2: same as in Model 1 and fear of childbirth in 2nd pregnancy, birth experience after 1st birth and mode of delivery in 1st birth (elective vs unplanned CD)

[b]ERCD (elective repeat cesarean delivery), VBAC (vaginal birth after cesarean), URCD (unplanned repeat cesarean delivery)

## Strengths and limitations

The major strength of this study was the population-based design with access to prospectively collected data in standardized electronic medical records. A population-based study design strengthens the external validity and it´s generalizability through less selection bias. All the information on maternal characteristics, pregnancy and delivery outcome was recorded before the outcome of this study, minimizing the risk of both selection and recall bias. As many as 69% of all women performed a TOLAC and 70% of them succeeded with a VBAC, which is in line with previous studies, strengthening the consistency and generalizability of our study results [29, 30].

VAS score is an accessible, easy and understandable tool, today still used in the clinics to evaluate overall birth experience measured a few days after childbirth. VAS is a valid prediction instrument of birth experience and, as a simple alternative, have a high correlation with other birth experience scales such as Wijma Delivery Experience Questionnaire B and is shown to have a persistence over time [8, 31]. We therefore assume that the likelihood of measurement bias is low when using the VAS score in this study. However, birth experience may in each woman entail a variety of feelings and experiences in many different dimensions [5], thus, it is important to be aware of the limitations of this scale as a tool for deeper understanding of the birth experience. Additionally, the question asked about birth experience is not standardized and non-anonymous, as it is asked by the midwife responsible for the postnatal care. Women may be intimidated or hesitant to answer the question honestly in the presence of a midwife with the tendency to bias the response toward a positive experience. There is also a

**Table 3. Odds of negative birth experience by mode of delivery in 2nd birth, logistic regression.**

| Mode of delivery in 2nd birth[b] | Negative birth experience 2nd birth | | | | | | | | |
|---|---|---|---|---|---|---|---|---|---|
| | Crude | | | | Model 1[a] | | Model 2[a] | |
| | n | % | OR | 95% CI | aOR | 95% CI | aOR | 95% CI |
| ERCD | 8 | 3.2 | Reference | | Reference | | Reference | |
| VBAC | 46 | 11.9 | 4.1 | 1.9–8.8 | 3.8 | 1.6–9.3 | 2.2 | 0.7–7.2 |
| URCD | 29 | 17.2 | 6.3 | 2.8–14.1 | 6.2 | 2.4–15.8 | 5.0 | 1.5–16.5 |

[a]Adjustment in

Model 1: maternal age, height, BMI, cohabiting, education, self-assessed health in 2nd pregnancy

Model 2: same as in Model 1 and fear of childbirth in 2nd pregnancy, birth experience after 1st birth and mode of delivery in 1st birth (elective vs unplanned CD)

[b]ERCD (elective repeat cesarean delivery), VBAC (vaginal birth after cesarean), URCD (unplanned repeat cesarean delivery)

**Table 4. Crude mean difference of birth experience and crude odds of negative birth experience by mode of delivery in 1st and 2nd birth.**

| Mode of delivery in 1st and 2nd birth | Birth experience 2nd birth | | | | Negative birth experience in 2nd birth | | | |
|---|---|---|---|---|---|---|---|---|
| | n | Mean (SD) | β | 95% CI | n | % | Crude OR | 95% CI |
| **Elective CD in 1st birth, 2nd birth[a]** | | | | | | | | |
| ERCD (Reference) | 96 | 8.8 (1.4) | | Reference | 3 | 3.1 | | Reference |
| VBAC | 98 | 8.1 (1.8) | -0.7 | -1.2 to -0.2 | 10 | 10.2 | 3.5 | 0.9–13.2 |
| URCD | 21 | 7.3 (2.4) | -1.5 | -2.4 to -0.6 | 4 | 19.1 | 7.3 | 1.5–35.5 |
| **Unplanned CD in 1st birth, 2nd birth[a]** | | | | | | | | |
| ERCD | 155 | 8.8 (1.4) | 0.0 | -0.5 to 0.5 | 5 | 3.2 | 1.0 | 0.2–4.4 |
| VBAC | 290 | 8.0 (2.1) | -0.8 | -1.2 to -0.4 | 36 | 12.4 | 4.4 | 1.3–14.6 |
| URCD | 148 | 7.7 (2.1) | -1.1 | -1.6 to -0.6 | 25 | 16.9 | 6.3 | 1.8–21.5 |

[a]ERCD (elective repeat cesarean delivery), VBAC (vaginal birth after cesarean), URCD (unplanned repeat cesarean delivery)

risk that when asked about experience shortly after the birth the women might be affected by the so called halo-effect. Evidence shows that women rate their experience more positive shortly after birth compared with ratings later [5]. However, this halo effect most likely affects all women regardless of mode of delivery.

This study has other limitations. We included 12 of the 42 hospitals in the country based on if they had a VAS response rate of 80% or higher. The 12 hospitals may not be representative of all Swedish hospitals, but including hospitals with low VAS response rate could introduce more selection bias on individual level e.g. low response hospitals may only ask women when birth experience was presumably negative. Our sensitivity analysis did however show that the mean VAS scores where about the same when including all possible hospitals (excluding the southeast region). Women giving birth in the included hospitals but having missing VAS had similar maternal characteristics and birth outcome as those included in the study population. We therefore conclude that selection to be scored with VAS appeared to be at random in the included hospitals. Nevertheless, there is always a risk of residual confounding. Additionally, the study period is short which limits statistical power.

## Interpretations

This study confirms that birth experience is associated with mode of delivery in women with a previous CD. This finding is in line with previous studies based on intervention programs with education of the women and decision-aids for counselling programs, aiming at improving birth experience [22–25]. However, these studies are mostly with small and clinical-based samples, subject to selection bias in opt-in or loss to follow-up, or with unadjusted confounding of parity or gestational age [22–24]. We found a slightly lower mean birth experience among women with a VBAC, which is different from Cleary Goldman et al. who found that women with a VBAC were most satisfied [22]. This difference in comparison with our results may be explained by the intervention program conducted in the study by Cleary Goldman et al. motivating women to perform a TOLAC [22]. In Sweden, women with one previous cesarean delivery are recommended to undergo TOLAC if there are no contraindications for vaginal delivery. Women discuss mode of delivery with the antenatal midwife and she is encouraged to a trial of labor. If the woman is hesitant about trial of labor, she will be referred to an obstetrician. Obstetricians in Sweden tend to be generous allowing women with one previous cesarean having an ERCD upon request. Therefore, an explanation for general high birth experiences regardless of delivery mode could be the shared decision and good quality of care. Shorten et al. also found higher VAS score among VBAC women, examined 6–8 weeks after birth [23, 25].

Women with a VBAC in our sample might have had higher scores if they had been measured later after birth. However there are studies showing the consistency of a measured birth experience even after a certain time has passed [31]. Shorten et al. also had a great difference in successful VBAC rates in the different sites (48% vs 74%, expected VBAC rates are 60–80% [19, 20, 29]), reducing its generalizability [23]. Cleary Goldman et al. had a large loss of women, only enrolling 95 women of 316 eligible, disposable for selection bias and also including premature births probably affecting birth experience [22]. When including women with mixed parity and mixing preterm and term births there is a possibility of diluting the results.

By excluding women with limited ability to speak or understand English as in the study by Emmet et al. the generalizability decreases [24]. Furthermore, women lost to follow-up were younger and had higher deprivation scores, increasing risk of selection bias depending on socio-economic status [24]. Emmet et al. showed that women with an unplanned repeat CD had the lowest VAS score (mean 48.5, scale 0–100), [24] which is in line with our results. However the mean rating was very low in comparison with our birth experience outcome, possibly explained by a measurement context bias with the VAS scale; the context bias is when the scale has many better or worse states presented in the scale and therefore the values may be depressed or enhanced due to cognitive processes used by the respondents [32].

After an intervention, as in all three above mentioned studies, the increasing knowledge and awareness due to the intervention may not reflect the birth experience of the general population, introducing bias and less generalizability.

This study shows that it is of importance not only to take into account the risk of medical and physical outcomes in women with a previous CD but also acting on the risk of negative birth experience, since this can have long-term implications on the woman and her family [2–4]. An unplanned CD is associated with negative birth experience in the 1[st] birth, confirming that this group of women are a more sensitive group [5]. We also indicate that whether the 1[st] cesarean was planned or not may influence the birth experience in the 2[nd] birth, in women with a first elective CD and a following unplanned CD possibly reflecting the disappointment of a first failed attempt of vaginal birth. Nonetheless, this study also supports TOLAC, since the vast majority of women giving birth after cesarean rate their birth as positive regardless of mode of delivery.

After one previous cesarean most women in Sweden are recommended and encouraged to go through a TOLAC, without attending any special program of care during pregnancy. During labor, women with a previous CD are considered having higher risk and are medically monitored more closely than low risk women. We suggest that women with a previous CD may benefit from an individual care plan throughout pregnancy and delivery. After childbirth, they ought to be screened for negative birth experience and offered counselling. We need to increase the awareness and skills among caregivers, with the aim of decreasing adverse outcomes and improve birth experience. More studies are needed in providing better support and maternity care.

## Conclusion

Most women with a previous cesarean scored their birth experience as positive independently of second mode of delivery. However, women performing a trial of labor ending up with a repeat unplanned cesarean had a five-fold higher odds of negative birth experience in comparison with women having elective repeat cesarean. Women with a previous cesarean, are at risk of adverse outcomes and may benefit from special attention, care and support before and during delivery. After delivery, women with an unplanned second CD should be screened for birth experience and offered counselling when needed.

## Supporting information

**S1 Table. Maternal characteristics by planned mode of delivery in 2<sup>nd</sup> birth.**
(DOCX)

**S2 Table. Mean difference of birth experience by planned mode of delivery in 2<sup>nd</sup> birth, linear regression.**
(DOCX)

**S3 Table. Odds of negative birth experience by planned mode of delivery in 2<sup>nd</sup> birth, logistic regression.**
(DOCX)

**S4 Table. Maternal characteristics by registered birth experience in 2<sup>nd</sup> birth.**
(DOCX)

## Author Contributions

**Conceptualization:** Charlotte Lindblad Wollmann, Can Liu, Mia Ahlberg, Olof Stephansson.

**Data curation:** Charlotte Lindblad Wollmann.

**Formal analysis:** Charlotte Lindblad Wollmann, Can Liu.

**Funding acquisition:** Olof Stephansson.

**Resources:** Olof Stephansson.

**Supervision:** Olof Stephansson.

**Writing – original draft:** Charlotte Lindblad Wollmann.

**Writing – review & editing:** Charlotte Lindblad Wollmann, Can Liu, Sissel Saltvedt, Charlotte Elvander, Mia Ahlberg, Olof Stephansson.

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
