## [Decision Letter · Decision Letter 0]

26 Nov 2019

PONE-D-19-19552

Risk of negative birth experience in trial of labor after cesarean delivery: a population-based cohort study

PLOS ONE

Dear Dr Lindblad Wollmann,

Thank you for submitting your manuscript to PLOS ONE. After careful consideration, we feel that it has merit but does not fully meet PLOS ONE’s publication criteria as it currently stands. Therefore, we invite you to submit a revised version of the manuscript that addresses the points raised during the review process.

ACADEMIC EDITOR: This article aims to investigate the birth experience related to the mode of de delivery after a previous cesarean section using data from a population-based cohort of women registered in the Swedish Pregnancy register. The topic of this study is of interest, whereas there are only few data in the literature. The article is well written and provide relevant information. But, some major revision should be done.

First, as noted by the two reviewers, the authors state that they included patients with a first cesarean and eligible for TOLAC. However, they don’t give any information in the methods section about what they consider being eligible for TOLAC. The authors should also provide details on the parameters which have leaded to the decision of TOLAC or ERCD.

Second, as noted by reviewer 1, another important point is that it would be relevant to compare the birth experience regarding the intent mode of delivery, not only the actual mode of delivery, i.e. TOLAC vs ERCD, not only VBAC vs ERCD vs URCD.

Third, as noted by reviewer 2, major revision should be done to precise some results.

We would appreciate receiving your revised manuscript by Jan 10 2020 11:59PM. To enhance the reproducibility of your results, we recommend that if applicable you deposit your laboratory protocols in protocols.io, where a protocol can be assigned its own identifier (DOI) such that it can be cited independently in the future. For instructions see: http://journals.plos.org/plosone/s/submission-guidelines#loc-laboratory-protocols

We look forward to receiving your revised manuscript.

Kind regards,

Guillaume Ducarme, MD, MSc, PhD

Academic Editor

PLOS ONE

Journal Requirements:

3. In ethics statement in the manuscript and in the online submission form, please provide additional information about the database used in your retrospective study. Specifically, please ensure that you have discussed whether all data were fully anonymized before you accessed them and/or whether the IRB or ethics committee waived the requirement for informed consent. If patients provided informed written consent to have their data used in research, please include this information.

4. Your ethics statement must appear in the Methods section of your manuscript. If your ethics statement is written in any section besides the Methods, please move it to the Methods section and delete it from any other section. Please also ensure that your ethics statement is included in your manuscript, as the ethics section of your online submission will not be published alongside your manuscript.

Reviewers' comments:

Reviewer's Responses to Questions

**Comments to the Author**

1. Is the manuscript technically sound, and do the data support the conclusions?

Reviewer #1: Yes

Reviewer #2: Yes

2. Has the statistical analysis been performed appropriately and rigorously? 

Reviewer #1: Yes

Reviewer #2: Yes

3. Have the authors made all data underlying the findings in their manuscript fully available?

Reviewer #1: No

Reviewer #2: Yes

4. Is the manuscript presented in an intelligible fashion and written in standard English?

Reviewer #1: Yes

Reviewer #2: Yes

5. Review Comments to the Author

Reviewer #1: This article aims to investigate the birth experience related to the mode of de delivery after a previous cesarean section.

It’s an interesting point whereas there are only few data in the literature. The article is well written and provide relevant information.

Here are several comments:

In the abstract, the authors state that they included patients with a first cesarean and eligible for TOLAC. However, they don’t give any information in the methods section about what they consider being eligible for TOLAC. Indeed not every patients with a previous cesarean section are eligible for TOLAC, depending on different surgical and obstetrical parameters.

The authors should provide details on the parameters which have leaded to the decision of TOLAC or ERCD. Did the patients have always the choice? In this kind of study, choice of the patients is a key-point since we can hypothesize that the birth experience might be better when the patients could actually choose the mode of delivery.

Another important point is that it would be relevant to compare the birth experience regarding the intent mode of delivery, not only the actual mode of delivery, i.e. TOLAC vs ERCD, not only VBAC vs ERCD vs URCD. Indeed, at the time we counsel patients, we don’t know how TOLAC will work…

VAS is a very poor tool to assess a complex feeling as birth experience. Even if the authors acknowledge limits of VAS in the discussion section, it should be more pointed out.

The figures (flow-chart and histograms) are of poor quality.

Reviewer #2: General comment:

The article shows data from a population-based cohort of women registered in the Swedish Pregnancy register. The topic of this study is of interest. As described by the authors, few publications focus on women's experience in second birth among women with a first cesarean. I found the manuscript well written overall.

Abstract

The abstract is clear. Minor comments: you may detail the number of women and frequencies in each categories for the exposure in the results section( n(%) for ERCD , URCD, VBAC). I would prefer using the terms of ‘odds’ instead of ‘risk’ when saying ‘URCD had five-fold …etc’.

Introduction

Very clear, thank you.

Methods

Please be more precise, if possible, when describing how women usually report their experience: is the scoring is self-administered or is the question asked by / in presence of the midwife? Are the answers anonymous? It could be a point of discussion: women may be intimidated to answer the question honestly in the presence of the caregiver tending to bias the responses toward a positive experience. Is the question about childbirth’s experience standardized across maternity units?

What do you mean by ‘organizational factors’ that could have an effect on the rate of reporting birth experience? Was the mean birth experience in these maternity units with birth experience response rate of less than 80% different from those with higher response rate ? Why choosing the cut off of 80% for exclusion?

Please consider rephrasing the sentences explaining the cut-off of 5 for defining negative birth experience. Here is a suggestion (line 130-131): “Previous studies … […]. The tenth percentile of the distribution of birth experience was 5 in our study. Thereby, to find women …etc”.

Do you have any information wether the decision of TOLAC or ERCD was shared with the women or decided only by the medical team? Many studies dealing with birth experience demonstrate the importance of shared-decision making process with the women; and that feeling of not having the choice was associated with lower satisfaction and experience, whatever the outcome.

Do you also have information on why ERCD was performed instead of TOLAC ? Is there clinical factors that systematically contra-indicate TOLAC and are those standardized across hospitals? Was the rate of TOLAC homogeneous across hospitals?

Statistical analysis

Before performing Student t test and linear regression, have you checked that hypothesizing a normal distribution seemed correct? Please consider justifying that somewhere.

Results

The Table 1 is not very clear and in line with your method section. As your exposure variable is 1) ECRD ; 2) unplanned repeated CD ; 3) VBAC, I recommend not to compare ERCD with TOLAC but to compare ERCD with 2) and 3) (two-by-two), as described in the method. It will also be more in line with the way you analysed them in the multivariable models. The rate of TOLAC may just be detailed in the text. Perhaps the differences in women’s characteristics between ERCD and TOLAC might be added as supplemental data ?

Line 178: the step-wise method for adjustment should be introduce in the method section and not in the results.

Table 2: it’s not very clear how you selected the confounders for mode l1 and model 2. Please better explain that in the method section.

Table 4: I’m not sure to clearly understand the table. What is the category of reference for the strata ‘unplanned CD in 1st birth, 2nd birth’? As far as I know, stratifying the results imply that the analyses of the effect of exposure on outcome are performed separately in each strata of a confounder (here elective or unplanned CD in first birth). Shouldn't the reference in stratum 2 be ERCD (but among women with 1st unplanned CD) as well?

Line 198-9: I’m not sure you can conclude so strongly of a higher risk of negative second birth experience for women with unplanned CD in 1st birth given the very small number of women and the very wide confidence interval. Perhaps moderate the sentence with ‘it seems that …’?

Table S1: it may be interesting to describe difference in hospital characteristics among women with and without missing data on childbirth experience (information as private maternity status and university status).

Discussion

Line 18-19: same comment as above.

Line 240-242: Couldn’t this assumption be easily verified?: were the scores of birth experience lower in hospitals were the scoring rate was lower than 80%?

Line 249: it’s not clear on which studies you referred to: what kind of intervention programs do you mean? We understand it later in the section but detailed it briefly here for a better understanding.

As commented before, I suggest that you discuss more about the importance of share decision making process and women’s involvement in this. It may be an element of explanation as to why the experience is high whatever the outcome of delivery.

You also introduce the importance of postpartum counselling in the conclusion. It’s an important element of discussion that, in my view, merit to be mention in the discussion section.

6. PLOS authors have the option to publish the peer review history of their article (what does this mean?). If published, this will include your full peer review and any attached files.

Reviewer #1: No

Reviewer #2: Yes: Blanc-Petitjean Pauline

---

## [Author Response · Author response to Decision Letter 0]

10 Jan 2020

We have performed major revisions and answered the questions from the reviewers. We have incorporated all your suggestions into our revision. We are very thankful of your help.

---

## [Decision Letter · Decision Letter 1]

4 Feb 2020

Risk of negative birth experience in trial of labor after cesarean delivery: a population-based cohort study

PONE-D-19-19552R1

Dear Dr. Lindblad Wollmann,

We are pleased to inform you that your manuscript has been judged scientifically suitable for publication and will be formally accepted for publication once it complies with all outstanding technical requirements.

With kind regards,

Guillaume Ducarme, MD, MSc, PhD

Academic Editor

PLOS ONE

Additional Editor Comments (optional): All comments have been addressed. As noted by Reviewer 2 (Pauline Blanc-Petitjean), substantial modifications have been made by the authors, the answers are relevant, and presentation of the results is more clear.

Reviewers' comments:

Reviewer's Responses to Questions

**Comments to the Author**

1. If the authors have adequately addressed your comments raised in a previous round of review and you feel that this manuscript is now acceptable for publication, you may indicate that here to bypass the “Comments to the Author” section, enter your conflict of interest statement in the “Confidential to Editor” section, and submit your "Accept" recommendation.

Reviewer #1: All comments have been addressed

Reviewer #2: All comments have been addressed

2. Is the manuscript technically sound, and do the data support the conclusions?

Reviewer #1: Yes

Reviewer #2: Yes

3. Has the statistical analysis been performed appropriately and rigorously? 

Reviewer #1: Yes

Reviewer #2: Yes

4. Have the authors made all data underlying the findings in their manuscript fully available?

Reviewer #1: Yes

Reviewer #2: Yes

5. Is the manuscript presented in an intelligible fashion and written in standard English?

Reviewer #1: Yes

Reviewer #2: Yes

6. Review Comments to the Author

Reviewer #1: (No Response)

Reviewer #2: Substantial modifications have been made by the authors and the answers are relevant. Presentation of the results is more clear. Thank you.

7. PLOS authors have the option to publish the peer review history of their article (what does this mean?). If published, this will include your full peer review and any attached files.

Reviewer #1: No

Reviewer #2: Yes: Pauline Blanc-Petitjean

---

## [Editor Report · Acceptance letter]

18 Feb 2020

PONE-D-19-19552R1 

Risk of negative birth experience in trial of labor after cesarean delivery: a population-based cohort study 

Dear Dr. Lindblad Wollmann:

I am pleased to inform you that your manuscript has been deemed suitable for publication in PLOS ONE. Congratulations! Your manuscript is now with our production department. 

With kind regards,

on behalf of

Dr. Guillaume Ducarme 

Academic Editor

PLOS ONE